# Extracellular DNA: A Critical Aspect of Marine Biofilms

**DOI:** 10.3390/microorganisms10071285

**Published:** 2022-06-24

**Authors:** Benjamin Tuck, Silvia J. Salgar-Chaparro, Elizabeth Watkin, Anthony Somers, Maria Forsyth, Laura L. Machuca

**Affiliations:** 1Curtin Corrosion Centre, WA School of Mines: Minerals, Energy and Chemical Engineering, Curtin University, Kent Street, Bentley, WA 6102, Australia; benjamin.tuck@postgrad.curtin.edu.au (B.T.); silvia.salgar@curtin.edu.au (S.J.S.-C.); 2Curtin Medical School, Curtin University, Kent Street, Bentley, WA 6102, Australia; e.watkin@curtin.edu.au; 3Institute for Frontier Materials, Deakin University, Geelong, VIC 3217, Australia; anthony.somers@deakin.edu.au (A.S.); maria.forsyth@deakin.edu.au (M.F.)

**Keywords:** extracellular DNA, microbiologically influenced corrosion, biofilm, extracellular polymeric substances, EPSs

## Abstract

Multispecies biofilms represent a pervasive threat to marine-based industry, resulting in USD billions in annual losses through biofouling and microbiologically influenced corrosion (MIC). Biocides, the primary line of defence against marine biofilms, now face efficacy and toxicity challenges as chemical tolerance by microorganisms increases. A lack of fundamental understanding of species and EPS composition in marine biofilms remains a bottleneck for the development of effective, target-specific biocides with lower environmental impact. In the present study, marine biofilms are developed on steel with three bacterial isolates to evaluate the composition of the EPSs (extracellular polymeric substances) and population dynamics. Confocal laser scanning microscopy, scanning electron microscopy, and fluorimetry revealed that extracellular DNA (eDNA) was a critical structural component of the biofilms. Parallel population analysis indicated that all three strains were active members of the biofilm community. However, eDNA composition did not correlate with strain abundance or activity. The results of the EPS composition analysis and population analysis reveal that biofilms in marine conditions can be stable, well-defined communities, with enabling populations that shape the EPSs. Under marine conditions, eDNA is a critical EPS component of the biofilm and represents a promising target for the enhancement of biocide specificity against these populations.

## 1. Introduction

Biofilms are a pervasive threat to marine infrastructure. The impacts of biofilm formation on metals and other materials manifest as biofouling, contamination, and microbiologically influenced corrosion (MIC) [1,2,3,4]. MIC alone can be expected to contribute 20–30% of all global corrosion costs, amounting to a conservative USD 30–50 billion per annum [5,6]. Biofouling and MIC are not well understood or effectively controlled in the marine environment, leading to the application of toxic, broad-spectrum chemical treatments (biocides). Biocides represent a primary line of defence against biofilms on marine infrastructure. 

To remain effective against adaptive microbial populations, regular biocide optimisation is required [7,8]. However, the fundamental understanding of natural marine biofilm composition remains a bottleneck for biocide efficacy improvement. Species diversity represents a contemporary challenge for natural biofilm research. Multispecies biofilms host complex behaviour, which single-species simulations fail to reproduce [9]. Competition and synergistic relationships within the community, for example, can help shape the molecular and species composition of the biofilm [10,11]. Scientific literature published on the extracellular and cellular composition of multispecies biofilms is limited, leading to the absence of low-toxicity, effective, and targeted biocide options [12]. Understanding fundamental aspects of biofilm formation will assist the management of deleterious biofilms into the future, especially as greener treatment options are becoming more desirable [13]. 

The EPS provides resident cells with a physical and chemical barrier reported to enhance biocide tolerance by as much as 1000 times compared to planktonic counterparts [14,15]. Extracellular polymeric substances (EPSs) are produced by bacterial cells to form the biofilm matrix [16]. The EPS composition has been screened across a variety of terrestrial and nonterrestrial environments, revealing polysaccharides, proteins, nucleic acids, and lipids as major constituents [17]. In marine multispecies biofilms developed on metal surfaces, the range and abundance of EPS components is relatively unexplored. In other aqueous environments (for example, food processing plants), the EPSs are considered primarily polysaccharides and proteins [18,19,20,21]; however, significant gaps in understanding are still evidenced by inadequate biofilm control measures. Specifically pertaining to engineered systems in seawater, the EPSs can also interact with iron and pose a direct MIC risk [22]. Since EPSs provide many functions linked to survival of the biofilm, including substrate attachment, protection, and horizontal gene transfer, understanding the EPS composition is critical for a targeted approach to biofilm mitigation [23]. 

Extracellular DNA (eDNA) is one biofilm matrix component gaining considerable traction in recent years due to its important structural role [24]. Over the past two decades, eDNA-degrading enzymes have been associated with biofilm dispersal [25,26], thus establishing nucleic acids as critical matrix polymers. Although a plethora of research identifies and describes the role of eDNA in the context of clinical biofilms [24,27,28], research relating to environmental biofilm EPS is limited. In this communication, marine biofilms developed on steel are hypothesised to produce eDNA and share a similar dependence on eDNA for structural integrity. 

Although further research is required to catalogue the EPS composition in marine biofilms, EPS production is closely associated with population dynamics. Even in single-species biofilms, genetic variants are exploited for upregulated EPS production pathways to promote the survival of the greater biofilm [29]. Similarly in dual-species biofilms, EPS quantity is strongly influenced by interactions between species [30]. Since changes in multispecies biofilm populations are likely to influence the biofilm tolerance profile, EPS research should be supplemented by information on the contributing biofilm species. 

The present research aimed to progress the understanding of marine multispecies biofilms on metal surfaces by: (A) investigating the prevalence of major EPS components, in particular, eDNA, in marine biofilms, (B) characterising population dynamics and their association with eDNA synthesis and composition, and (C) outlining the importance of eDNA in the biofilm developmental process. To address research objectives, marine microorganisms were allowed to form biofilms on carbon steel (CS) over a period of 6 weeks. Over the course of the study, genomic and eDNA were extracted and characterised from the biofilms to understand how community structure fluctuated over time and how the community composition impacted eDNA production. Additionally, RNA sequencing of the population was conducted to assess the link between activity and eDNA production. Finally, confocal laser scanning microscopy (CLSM), adenosine triphosphate (ATP) qualification, and viability assays (culture-based) were also included at each sampling period to confirm viability of the population. 

## 2. Materials and Methods

### 2.1. Microbial Isolates

Experiments were conducted using three marine bacterial strains with demonstrated ability to form biofilms on CS [31]. Before inoculation into experimental reactors, *Shewanella chilikensis* DC57 [32], *Pseudomonas balearica* EC28 [33], and a laboratory strain of *Klebsiella pneumoniae* were grown anaerobically in liquid phase cultures using ASW media (Appendix A) supplemented with Bacto™ casamino acids (3 g/L *w/v*), sodium pyruvate (3 g/L *w/v*), D (+) glucose (3 g/L *w/v*), and ammonium nitrate (NH_4_NO_3_; 3 g/L *w/v*). Strains in liquid phase cultures were counted using a Neubauer haemocytometer chamber and inoculated into reactors as previously described [31], using 10^5^ cells from each pure culture. 

### 2.2. Sample Preparation and Surface Finish

Coupons of 5 mm thickness with a working surface of 1.27 cm^2^ were cut from CS rods (AISI 1030). Coupons were prepared by wet grinding with a successively finer grit finish, in the order of 80, 120, and 320 (SiC grit paper), before electrocoating with Powercron^®^ 600 CX solution. The final working surface was freshly wet-ground to a circular 120 grit finish. The coupons were washed in 100% pure ethanol, dried under nitrogen gas, and irradiated with ultraviolet (UV) light using a biosafety cabinet for 10 min each side to sterilise. Coupons were inserted into Center for Disease Control (CDC) reactor (Biosurface Technologies, Bozeman, MT, USA) rods using a biosafety cabinet (aseptic conditions), UV irradiating the coupons again after manual manipulation.

### 2.3. Experimental Setup

Biofilms were developed in CDC reactors over 6 weeks as experimental replicates (Figure 1). ASW media (500 mL) were used in all bioreactor experiments as previously described [34], with the following modifications (Appendix A): solution 1 addition of glucose (0.9 g/L *w/v*), sodium pyruvate (0.55 g/L *w/v*), Bacto™ casamino acids (1.5 g/L *w/v*) to CaCl_2_ (0.1 g/L *w/v*), and NH_4_NO_3_ (1.2 g/L *w*/*v*). Experimental reactors were established simultaneously and flushed with pure nitrogen gas before inserting rods containing UV-treated coupons. Once anaerobic conditions were established, reactors were directly inoculated with equal cell numbers of each bacterial strain. The reactor solution was then maintained for the experiment duration at 30 °C under a constant nitrogen flow (90 mL/min) and agitation at 50 rpm. A reservoir containing reactor solution was connected to a peristaltic pump and calibrated to flush 30% of the reactor solution every 7 days. To allow the population to establish a biofilm, reactors remained under batch conditions until day 3 after inoculation (when turbidity was observed). Continuous flow conditions ensured a constant, limited nutrient availability. Sampling was conducted after 2, 4, and 6 weeks.

### 2.4. Confocal Laser Scanning Microscopy (CLSM)

CLSM was used to identify biofilm EPS components, directly monitor eDNA presence over the experiment, and visualise the viability of the biofilm. All CLSM analyses were conducted on a Nikon A1+ confocal microscope equipped with a 20× dry objective lens, using version 5.20 of Nikon NIS Elements software. Coupons were removed from reactors and lightly rinsed in phosphate-buffered saline (PBS; Sigma, pH 7.4, St. Louis, MO, USA) before staining.

Biofilm EPS components were targeted using the following stain and stain–lectin conjugate concentrations, optimised for biofilm samples: proteins were targeted using Sypro^®^ Orange (Thermo Fisher, Waltham, MA, USA) in a 5X concentration. DiYO™-1 (AAT Bioquest Inc., San Francisco, CA, USA) was used to visualise eDNA at a working concentration of 5 µM. Total polysaccharides were captured using Wheat Germ Agglutinin (WGA)–Alexa Fluor™ 633 conjugates (Thermo Fisher) and Concanavalin A (ConA)–Alexa Fluor™ 633 conjugates (Thermo Fisher) at 50 µg/mL and 100 µg/mL working concentrations, respectively. WGA and ConA conjugates were applied simultaneously to bind sialic acid and N-acetylglucosaminyl residues (WGA) as well as α-mannopyranosyl and α-glucopyranosyl residues (ConA). Stains were combined in Ultrapure milliQ water and applied to coupon surfaces for at least 10 min before rinsing lightly in PBS (Sigma, pH 7.4). Coupons were then transferred into a purpose-built dish for all CLSM (ibidi^®^, Gräfelfing, Germany). The dish contained a central hole of radius 10 mm, covered by a glass coverslip. This design preserved the biofilm architecture by preventing compression of the sample. 

After confirming the relative abundance of eDNA in biofilm EPSs, eDNA was stained independently on coupons using DiYO™-1 (AAT Bioquest). The stain was applied for 10 min at a working concentration of 5 µM before gently rinsing again in PBS (Sigma, pH 7.4) for eDNA visualisation. Independent CLSM analysis with DiYO-1 provided technical replicates to confirm eDNA presence over 6 weeks in experimental replicates, while confirming the absence of signal bleed-through in EPS staining protocols. Micrographs were captured sequentially using a 489.3 nm laser and a 500–550 nm emission filter. All microscope and software settings remained uniform between sampling times and micrographs, with the following exception: EPS micrographs were captured at a smaller resolution (512 × 512) to minimise cell death and changes to EPS as a result of longer acquisition time.

### 2.5. Scanning Electron Microscopy (SEM)

SEM sample preparation was conducted as previously described [31,35]. Briefly, coupons were removed from reactors and lightly rinsed in PBS (Sigma, pH 7.4) before fixing in 2.5% glutaraldehyde solution for 22 h at 4 °C. Fixed biofilms were dried overnight under pure nitrogen gas and sputter-coated with 9 nm of platinum before imaging on a Tescan MIRA variable pressure field emission scanning electron microscope (VP-FESEM). 

### 2.6. eDNA Extraction and Quantification

eDNA was quantified in the biofilm matrix using a Qubit fluorimeter and HS reagent kit (Thermo Fisher). Coupons with biofilms were removed from reactors and lightly rinsed in PBS (Sigma, pH 7.4) before transferring to tubes containing 2 mL of fresh PBS (Sigma, pH 7.4). To extract eDNA from biofilms, a basic digestion and filtration protocol was conducted. All cells and debris were removed from coupons using a nonlytic sonication procedure. Briefly, tubes containing coupons were vortexed for 30 s and sonicated for 10 s followed by 15 s on ice, repeating for 7 cycles. Large particles were then removed from the sample by centrifugation at 15,000× *g* for 5 min, and the supernatant was filtered using a Sartorius Minisart^®^ 0.2 µm pore polyethersulfone (PES) membrane filter. Fluorimetry was conducted directly on the filtered volume. This procedure was repeated for the reactor planktonic samples, without the detachment (homogenisation) stage.

### 2.7. Total DNA Extraction

Total DNA from planktonic and biofilm communities was extracted from duplicate experiments at 2, 4, and 6 weeks. Total biofilm and planktonic DNA samples contained genomic DNA and eDNA and are therefore referred to as total biofilm DNA and total solution DNA, respectively, throughout this communication. Total DNA was extracted using a DNeasy^®^ PowerSoil^®^ Pro DNA extraction kit (Qiagen, Hilden, Germany) following the manufacturer’s instructions. DNA from biofilm samples was extracted from the pellets after centrifugation at 15,000× *g* for 5 min described above. DNA from planktonic cells was extracted from pellets after centrifugation of 5 mL of the test solution at 15,000× *g* for 5 min. 

### 2.8. RNA Extraction

RNA from biofilm communities was extracted from biological replicate experiments at 2, 4, and 6 weeks to identify active populations within the biofilm over time. RNA was extracted using the RNeasy^®^ PowerBiofilm^®^ kit (Qiagen), as recommended by the manufacturer. Subsequently, RNA was treated with DNase using Turbo DNA-free kit (Invitrogen, Waltham, MA, USA) to remove the remaining DNA. A PCR targeting the 16S rRNA gene was performed to verify the complete removal of DNA. Afterwards, RNA was purified and converted to cDNA using a SuperScript IV first-strand synthesis system (Invitrogen).

### 2.9. 16S rRNA Sequencing and Data Analyses

eDNA and total DNA extracted from biofilms and solution and cDNA synthetised from RNA extracted from biofilms were used as a template to generate amplicons of the V3–V4 gene region of the bacterial 16S rRNA gene for the estimation of the relative abundance of each isolate in the community. PCR was conducted using the primers 341F (5′ CCTAYGGGRBGCASCAG 3′) and 806R (5′GGACTACNNGGGTATCTAAT 3′) [35]. PCR amplicons were sequenced on an Illumina MiSeq instrument with a V3 (600 cycles) kit (Illumina, San Diego, CA, USA).

Resulting sequences were processed using the Quantitative Insights Into Microbial Ecology (QIIME2) software pipeline (QIIME2 v. 2020.11) [36]. Raw reads were visually inspected with the demux plugin and quality filtered with DADA2 pipeline (–p-trunc-len-f = 280 and –p-trunc-len-r″ = 220) [37]. The DADA2 plugin was also used to denoise and obtain representative amplicon sequence variants (ASV). Representative sequences and their abundances were extracted by feature-table plugin [38] and taxonomically classified using the Naïve Bayesian classifier against the SILVA database v.138 [39]. 

In order to visualise the multivariate dispersion of the community composition based on the DNA source and the microbial community at each sampling period, a nonmetric multidimensional scaling (NMDS) analysis was conducted based on the weighted UniFrac distance matrix [40]. The NMDS was performed in Rstudio (v1.3.1093) [41] using the “vegan” R package [42]. Microbial taxa were fit into the ordination by using the envfit function, and their significance was assessed under 999 permutations. Correlation between the NMDS and the relative abundance of the microbial taxa was considered significant if *p*-value < 0.05.

A linear discriminant analysis (LDA) effect size (LEfSe) [43] was applied to identify the specific bacterial taxa significantly associated with DNA source (‘eDNA’ or ‘total biofilm DNA’) or with the microbial community (‘biofilm’ or ‘planktonic’). For LEfSe, Kruskal–Wallis and pairwise Wilcoxon tests were performed, followed by LDA to assess the effect size of each differentially abundant taxon. A *p*-value of <0.05 was considered significant for both statistical methods. The threshold for the logarithmic discriminant analysis (LDA) score was set to 3.

### 2.10. Colony Forming Unit (CFU) Quantification

CFUs were extracted from coupons in 10 mL PBS (Sigma, pH 7.4) using the nonlytic sonication and vortex procedure described above. CFU plates were prepared using ASW solution (Appendix A) with 3 g/L *w/v* Bacto™ casamino acids, 3 g/L *w/v* sodium pyruvate, 3 g/L *w/v* D (+) glucose, and 15 g/L *w/v* bacteriological agar and (Sigma). Ammonium nitrate (NH_4_NO_3_) was excluded since agar plates were prepared and cultivated in aerobic conditions where electron acceptor supplementation was not required. The drop plate method was then used to prepare and quantify CFUs according to existing standards [44].

### 2.11. Adenylate Energy Charge (AEC) Analysis

Adenosine triphosphate (ATP), adenosine diphosphate (ADP), and adenosine monophosphate (AMP) adenylates are critical for the metabolism of all cells [45,46]. Hydrolysis of phosphate in ATP forms the more energy-depleted ADP and AMP, releasing energy for use by cellular processes. The AXP assay kit takes advantage of cellular dependence on these molecules for a rapid, sensitive estimation of biofilm energy charge [45]. In the present research, AXP assays were conducted according to the manufacturer’s instructions, with an additional sonication and vortex stage as detailed above to facilitate detachment and lysis. The suspension was then processed through the AXP assay kit and Quench-Gone Organic Modified (QGO-M) ATP assay kit (LuminUltra Technologies, Ltd., Fredericton, NB, Canada). 

## 3. Results

### 3.1. Identification of eDNA as a Major Structural Polymer in the Biofilm

To determine the composition of the EPSs, biofilm components were targeted with specific stains and imaged using confocal laser scanning microscopy with eDNA presenting the greatest signal (Figure 2A–C). 

IMARIS (Bitplane) statistical analysis was conducted to determine the relative contribution of each macromolecule to the biofilm. After 2 weeks, eDNA contributed > 90% to the total biofilm composition (in terms of quantified signal abundance in comparison to other macromolecules screened) in experimental replicate 1 with proteins and polysaccharides contributing < 10% combined (Figure 2D). A similar trend was observed in experimental replicate 2 (Appendix A). Although some variation exists between the percent contribution of macromolecules, a dominant eDNA signal was consistent. 

Biofilm eDNA was micrographed separately in experimental replicates across 2, 4, and 6 weeks to confirm the abundance of this macromolecule over the relatively longer term. The results of the CLSM analysis in Figure 3A–F represent experimental replicate 1 (A–C) and experimental replicate 2 (D–F). The abundance of eDNA in biofilm samples collected over 6 weeks was comparable among all results as indicated by green fluorescence.

Scanning electron microscopy was conducted to understand the physical morphology of the biofilms after 2, 4, and 6 weeks. Structures resembling bacterial cells were observed at all time periods (examples indicated by arrows). The structure of the EPS, especially at earlier sampling periods (2–4 weeks), resembled a fibrous net-like appearance resembling eDNA. Two-week-old biofilms contain cell-like structures surrounded by an abundance of EPS (Figure 4). 

Microscopic data were supported by direct quantification of free-floating DNA in the biofilm and planktonic populations. At all the time points, the biofilm contained more eDNA than the solution with the greatest amount of eDNA observed at 4 weeks, with 190 ng/mL and 130 ng/mL observed in the biofilm and solution, respectively (Appendix A).

To assess the viability of the biofilm across the 6-week sampling period, live and dead CLSM assays, CFU quantification, and AXP assays were conducted. Viability was maintained for the duration of the experiment, although it appeared to decrease with sampling time. CLSM analysis revealed large, mushroom-like structures developed over 2 weeks (Figure 5A) that gradually reduced to thin homogeneous biofilms by six weeks (Figure 5C). Although CLSM indicated a reduced viability trend with time, live cells (green fluorescence) were still detected in all micrographs for the duration of the experiment. Viability was quantified to reveal a similar trend (gradual viability reduction). CFU counts revealed that the 2-week sampling period produced the greatest number of viable cells from coupons (Figure 6A), which corresponded to biofilm activity as indicated by AXP analysis (Figure 6B). Both techniques indicted a gradual decline in viability over 6 weeks. 

### 3.2. Community Composition (DNA-Based Sequencing)

Differences in the microbial community composition between the eDNA fraction and the total DNA in both biofilm (sessile) and reactor solutions were observed at all sampling points (Figure 7). A higher relative abundance of *K.*
*pneumoniae* in the biofilm total DNA was clearly evident, with similar contributions by the other two strains. After separation of the eDNA from the biofilm, DNA sequencing revealed that eDNA not associated with live cells was mostly produced by *P. balearica* and *S. chilikensis* (Figure 7). This trend was also observed in the solution, revealing again that *K. pneumoniae* eDNA contributed relatively little to the free eDNA pool than the other two strains. Additionally, the solution eDNA pool was significantly enriched by the *S. chilikensis* strain. Statistically significant biomarkers among groups were determined for use in the NMDS ordination analysis (Figure 8).

### 3.3. Biofilm Community Composition of Active Microorganisms

RNA-based sequencing profiles revealed that *Pseudomonas balearica* was the most active microorganism in the biofilm along the experimental period (Figure 9). A reduction in the relative abundance of *K. pneumoniae* and an increase in the relative abundance of the *Shewanella* genus were observed after 2 weeks of biofilm growth. Complementary DNA-based and RNA-based profiling indicate that the high relative abundance of the *Klebsiella* genus in biofilm total DNA fraction at all sampling periods was mainly related to dormant or inactive cells. 

## 4. Discussion

CLSM analysis targeted proteins, polysaccharides, and eDNA presence in the biofilm for visual representation and semiquantitative analysis. All macromolecules identified in this analysis have been associated with EPS in previous communications [47,48,49]. Although eDNA is often reported in biofilms, especially from clinical isolates, it is often not the primary EPS component [50,51]. Polysaccharides and proteins are more frequently identified and are believed to comprise the bulk of the EPSs in most environments [19,52,53,54]. CLSM results and postimage analysis results (Figure 2 and Appendix A) indicate that eDNA comprised the overwhelming majority of the EPSs under marine-simulating conditions presented in this study (>90% and >60% of fluorescent signal in replicates 1 and 2, respectively). This finding was supported by a separate CLSM analysis (Figure 3), which demonstrated an abundant fluorescence by the eDNA-specific stain DiYO-1™ in experimental replicates at 2, 4, and 6 weeks. Scanning electron microscopy (SEM) revealed a fibrous, net-like appearance in biofilms where eDNA was detected (Figure 4). Similar EPS structures have been reported in staphylococcus biofilms, where mesh structures were also associated with eDNA [55]. To support these microscopic observations consistent with the eDNA presence in the biofilm matrix, DNA quantification by fluorimetric analysis was conducted on biofilm and solution samples across a 6-week period (Appendix A). Fluorimetry results of biological replicates revealed the presence of eDNA peaked in the EPS and reactor solutions at four weeks, remaining above the 2-week values by the end of the experiments. 

The production of eDNA was evaluated on marine multispecies biofilms developed under oligotrophic conditions (low organic nutrient supply). CLSM (live/dead) assays, CFU quantification, and AXP analysis were conducted to ensure EPSs, and community analysis was performed on living (viable) biofilms. CLSM live/dead results (Figure 5) demonstrate that biofilm viability was maintained for the duration of the experiments, although biofilm architecture and live cells appeared to reduce with time. CFUs and adenylates AMP, ADP, and ATP were recovered and quantified from biofilm coupons to quantitatively estimate changes to biofilm viability, revealing a similar trend (reduced viability and cellular energy with time) across experimental replicates. Biofilm viability as determined by CFUs diminished over the experiment, consistent with CLSM findings (Figure 6). This was also supported by the AXP analysis (Figure 6), with available biofilm energy also reducing over the experiment. This behaviour is typical of oligotrophic conditions, which are known to exacerbate MIC. For example, in one MIC mechanism, biofilm starvation leads to metabolic pathway changes that can lead to electron extraction from metallic surfaces [56].

### 4.1. Origin of eDNA and Community Structure

To understand the dynamics of the sessile and planktonic populations and the origin of the eDNA, DNA-based and RNA-based sequencing were conducted. The results demonstrate that *Pseudomonas balearica* EC28, a laboratory strain of *K. pneumoniae* and *Shewanella chilikensis* DC57 constructed a reproducible multispecies biofilm in marine-simulating conditions over 6 weeks (Figure 7). The biofilm eDNA fraction was therefore expected to be the sum contribution of the relative abundance of the three community members; however, a relative abundance based on eDNA differed from the total DNA indicating that the contribution of each isolate to the eDNA fraction was not directly proportional to its concentration. 

The DNA-based sequencing analysis revealed unexpected differences between total DNA and eDNA samples as well as between the sessile and planktonic populations. For instance, the relative abundance of each strain varied between biofilm and solution samples (Figure 7). While *Klebsiella pneumoniae* dominated the total DNA as indicated by LefSe analysis (Appendix A), the abundance of *K. pneumoniae* in the eDNA fraction was significantly lower than the abundance of other strains. Therefore, *K. pneumoniae* contributed relatively little to the biofilm eDNA while contributing a large number of inactive or dormant cells to the population. This hypothesis is supported by the RNA sequencing results, revealing *K.*
*pneumoniae* genus was the least active (Figure 9). 

In the planktonic population, LefSe analysis revealed a statistically significant enrichment of *Shewanella* (Appendix A). Although T_0_ inoculations comprised relatively equal volumes of each community member, the final community composition established over time (probably in response to system parameters, such as interspecies interactions, temperature, attachment substrate, shear stress, and atmospheric conditions). Results of this investigation also imply that the planktonic community does not necessarily reflect the composition of the biofilm community. This was expected, since sessile and planktonic cell phenotypes can vary greatly [57,58], and bacteria colonise surfaces at various rates [31,58]. Finally, bacterial eDNA contributions are likely to be active or passive based on relative contributions by each strain. While a single strain can dominate the biofilm cellular complement, the results indicate that eDNA contribution can be produced predominantly by other strains in either the biofilm or surrounding solution. 

The biofilm RNA-based sequencing revealed community similarities at 3, 4, and 6 weeks (Figure 9). Relative biofilm contribution by abundance for the duration of the experiments was *Pseudomonas balearica > Shewanella > Klebsiella*. While DNA-based sequencing results demonstrated some variation in the biofilm and planktonic–cellular and eDNA contributions, RNA-based sequencing indicated a more stable biofilm structure. Indeed, RNA-based relative abundance is expected to be associated with the active fraction of the community. As expected, the stability of RNA-based compared to DNA-based diversity profiling results imply that eDNA persisted in the reactor solution longer than exogenous RNA. While RNA has been recovered from simulations up to 13 h after release to the environment, DNA can persist for years [59] and is generally expected to degrade at a slower rate due to a more stable double-helix structure compared to single-stranded RNA. In the present research, similarities between RNA analysis results (between replicates and at different sampling times) revealed that microorganisms in the local environment do not form a random community. Instead, each strain contributed to the community composition in a relatively reproducible way with a unique level of stable activity.

### 4.2. The Role of eDNA in Biofilms

In clinical and single-species biofilms, eDNA has been reported in the EPSs. For example, while polysaccharides and proteins were present, the EPS composition of *Pseudomonas aeruginosa* biofilms was reported to be primarily eDNA [60]. Indeed, the structural role of eDNA in *P. aeruginosa* biofilms is well characterised with distinct production pathways. *P. aeruginosa* can actively excrete eDNA or generate it through autolysis triggered by quorum-sensing events [61]. Sacrificing healthy cells for contribution of eDNA to the matrix indicated that eDNA plays a critical role in the matrix. In this species, eDNA forms a scaffold that provides structural stability [61,62,63]. In the present research, CLSM targeted the spatial distribution of eDNA, indicating a similar structural role in the biofilm under marine-simulating conditions. 

Interestingly, eDNA in the *Pseudomonas balearica* is also known to contribute to the tolerance of the biofilm to antimicrobials [63]. Wen-chi et al. associated exogenously supplemented eDNA to *P. aeruginosa* biofilms with tolerance to aminoglycosides [64]. Therefore, eDNA from the surrounding environment is incorporated into the biofilm to support structural integrity and enhance tolerance to chemical treatments. In the present study, *P. balearica* EC28 was included in the multispecies community. The strain was recently implicated in an MIC failure of an oil production facility in Western Australia [65]. After isolation and sequencing of the strain in previous work [33], attachment to steel was evaluated using SEM. In Appendix A, a pure culture of *P. balearica* EC28 was grown on CS over 24 h, revealing 100% surface coverage by the strain and net-like structures resembling an eDNA network. Microscopic screening of the strain indicated that *P. balearica EC28* may be central to biofilm eDNA contribution. Further analysis is required to identify the genetic pathways employed by this strain in either apoptotic or active excretion mechanisms. Although biofilm eDNA sequencing supports the hypothesis that *Pseudomonas* eDNA dominated the matrix, the results also imply that eDNA is contributed by *S. chilikensis* DC57 and *K. pneumoniae*. 

As with *Pseudomonas* sp., eDNA plays an important role in biofilms of *K. pneumoniae*. Recently, Liu et al. reported a novel biofilm structure designated the ‘R-biofilm’ in *K. pneumoniae*, formed by breaks to double-stranded DNA [66]. The ring-like structures contained proteins and eDNA. Importantly, the R-biofilm was implicated as the more protective phenotype against adverse conditions, such as chemical treatment. *K. pneumoniae* biofilms are characterised to a lesser extent compared to *P. aeruginosa*; however, the significance of further research in this area is related mainly to clinical impacts. 

Like *P. balearica* EC28, *S. chilikensis* DC57 is a true marine strain. The strain was cultivated and sequenced in a separate work after recovery from an MIC-related equipment failure [32,65]. Almost nothing is known about the EPS composition of *S. chilikensis* biofilms; however, eDNA sequencing results demonstrate that eDNA is contributed by this strain to the multispecies community. 

## 5. Conclusions

To effectively manage marine biofilms with a reduced environmental impact, a greater understanding of the EPSs and community composition is required. The present research aimed to enhance the understanding of natural biofilm EPSs by identifying the dominant structural matrix component under marine-simulating conditions. Subsequently, through DNA-based and RNA-based sequencing analysis, this communication aimed to underpin changes to population dynamics over time and assess the origin of eDNA in marine biofilms. Microscopic analysis, postimage analysis, and direct quantification of extracellular DNA (eDNA) suggest that eDNA is the most abundant and structurally important molecule in marine multispecies biofilms. Sequencing of eDNA, total biofilm DNA-based and RNA-based sequencing revealed that all originally inoculated bacterial strains contributed to the biofilm composition. The biofilm structure declined over time under limited, consistent metabolic supplementation, although cell viability and eDNA remained throughout the experiment. Interestingly, the active fraction of the biofilm determined by RNA-based sequencing revealed that relatively stable communities form on carbon steel over 6 weeks of exposure. Lastly, no correlation was found between activity within the biofilm and DNA presence in the system.

## Figures and Tables

**Figure 1 microorganisms-10-01285-f001:**
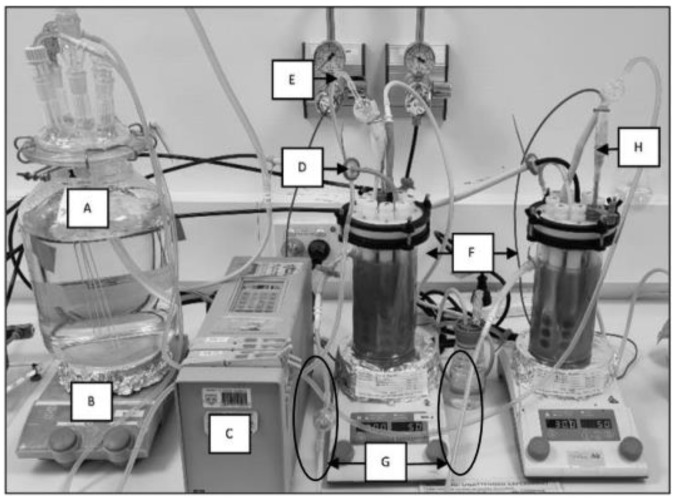
CDC reactor experimental set-up: (**A**) feeding cell with fresh reactor solution, (**B**) hot plate set to 30 °C and 50 rpm for CDC reactor, (**C**) pump for continuous flow replacing reactor solution by 30% weekly, (**D**) reactor gas inlet with 0.2 µm filter, (**E**) media inlet with air lock to prevent feeding cell contamination, (**F**) CDC reactor duplicate experiments, (**G**) reactor solution outlet for continuous flow (with air locks), and (**H**) thermocouple probe.

**Figure 2 microorganisms-10-01285-f002:**
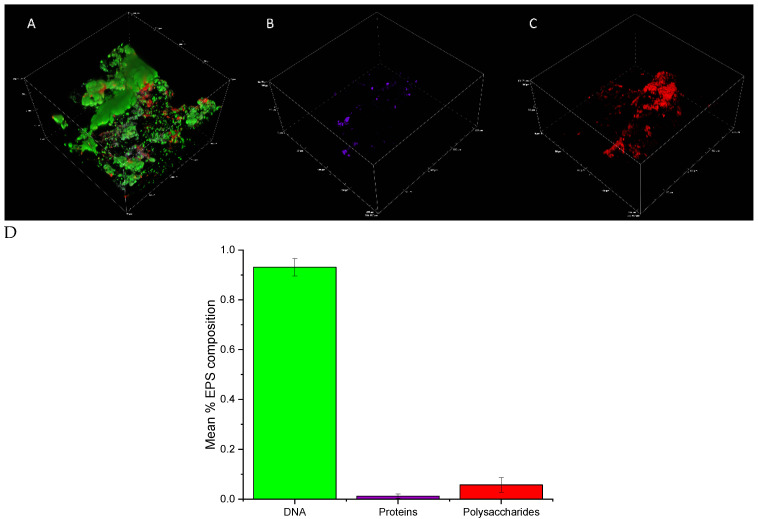
(**A**–**C**) Confocal micrographs of early biofilms (2 weeks) depicting the EPS composition, where (**A**) all channels combined, (**B**) protein-targeting channel, and (**C**) polysaccharide-targeting channel. (**D**) IMARIS (Bitplane) analysis depicting the average percent contributions of eDNA, proteins, and polysaccharides to the matrix of biofilms from experimental replicates. Error bars represent the standard deviation of triplicate micrographs.

**Figure 3 microorganisms-10-01285-f003:**
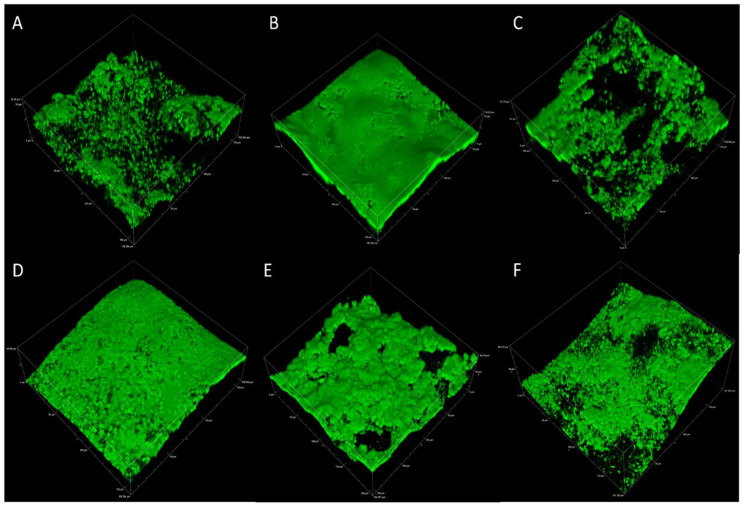
Representative CLSM of eDNA in multispecies biofilms after 2, 4, and 6 weeks for experimental replicate 1 (**A**–**C**) and experimental replicate 2 (**D**–**F**).

**Figure 4 microorganisms-10-01285-f004:**
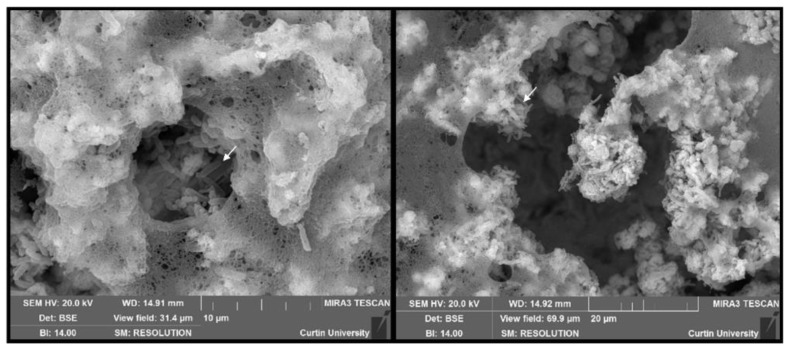
Scanning electron micrographs captured from 2-week-old biofilms showing a net-like structure within EPSs. Structures resembling bacterial cells are also evident in the samples.

**Figure 5 microorganisms-10-01285-f005:**
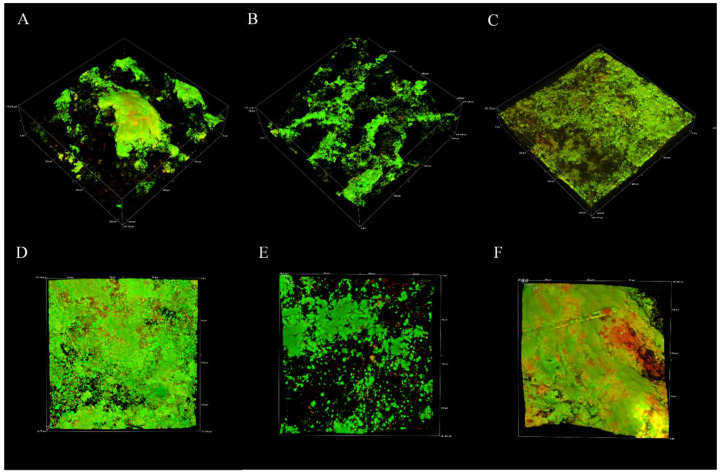
Live (green)and dead (red) staining of biofilms at 2, 4, and 6 weeks. Images were obtained with CLSM. (**A**–**C**) are larger micrographs (600 × 600 μm). (**D**–**F**) represent the same surface captured using the Nyquist function of the Nikon Elements software for increased resolution.

**Figure 6 microorganisms-10-01285-f006:**
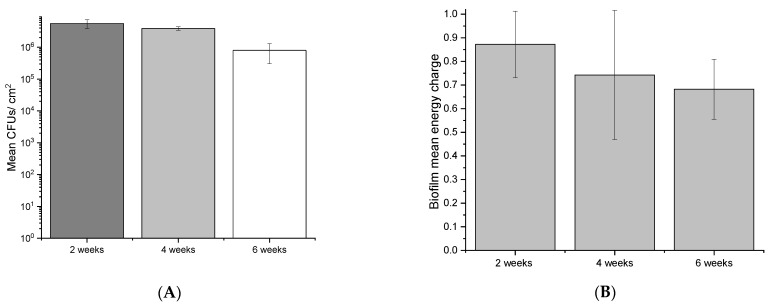
Mean CFUs (**A**) and the pooled energy charge (**B**) of biofilms at 2, 4, and 6 weeks. Error bars represent the standard deviation of 2 experimental and 3 technical replicates.

**Figure 7 microorganisms-10-01285-f007:**
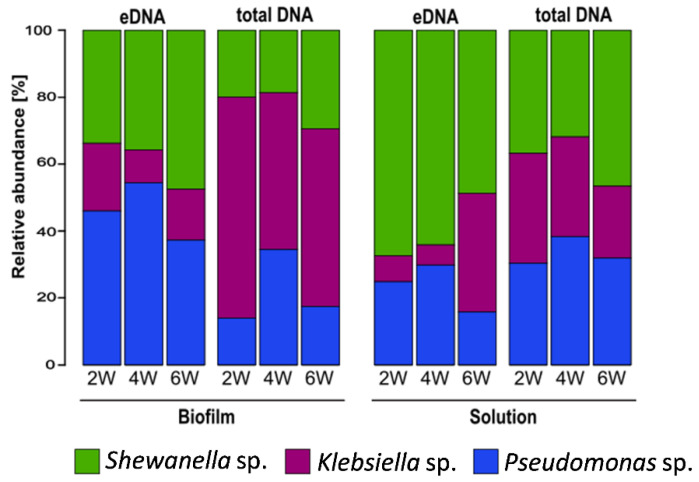
Mean relative abundance of biofilm and planktonic microbial taxa: eDNA: extracellular DNA. 2W: 2 weeks of exposure; 4W: 4 weeks of exposure; 6W: 6 weeks of exposure. Data are the average of two biological replicates.

**Figure 8 microorganisms-10-01285-f008:**
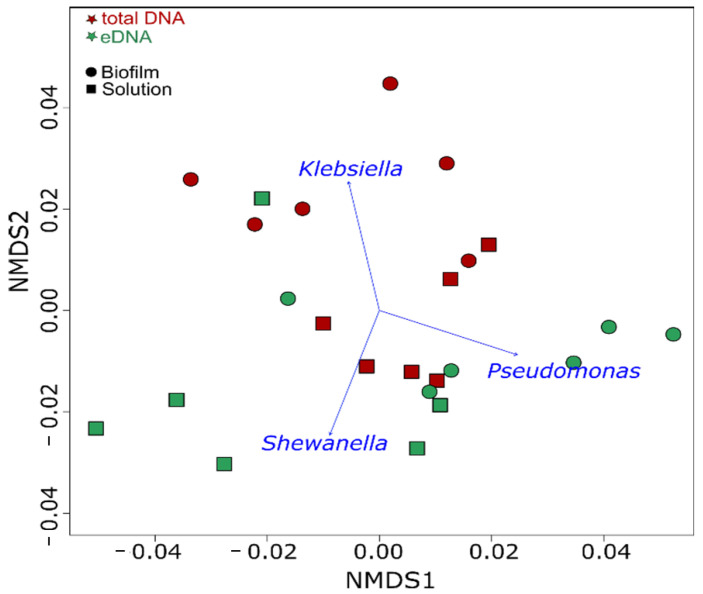
NMDS of the microbial communities at each sampling period. Microbial taxa significantly correlated (*p* = 0.001) with microbial community structure are indicated by blue arrows.

**Figure 9 microorganisms-10-01285-f009:**
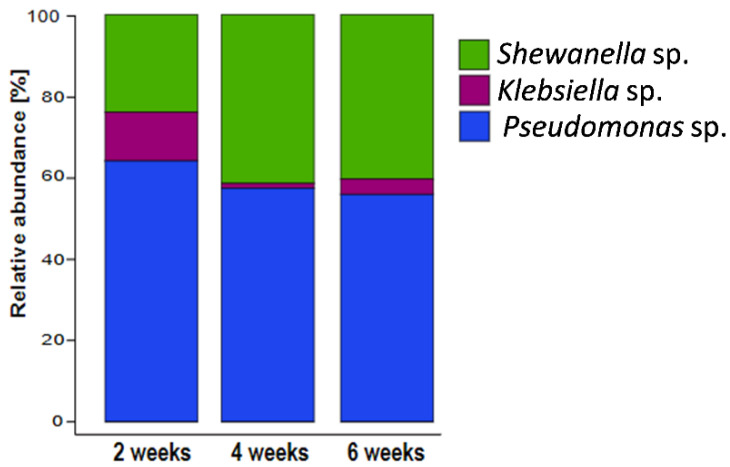
16S rRNA-based microbial community composition of biofilms showing the mean relative abundances of each microbial taxa. Data are derived from the average abundance of two technical and two experimental replicates.

## Data Availability

Not applicable.

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
