# Peer review of "Extracellular DNA: A Critical Aspect of Marine Biofilms"

_microorganisms, 2022, doi:10.3390/microorganisms10071285_

Round 1
Reviewer 1 Report
Address the following comments
1. There are several studies has been reported about biofilm metagenomics and their impact on microbial corrosion, how would differentiated this study?
2. Why three bacteria were used in this study, already this three bacterial strains tested with MIC characteristics such as Weight loss, impedance and polarization individually, In addition the three bacteria isolation origin where?
3. The bacteria f Klebsiella pneumoniae pathogenicity test was conducted? If already conducted what is the rate of pathogenicity.
4. During the biofilm development or after why not conducted Electrochemical study of the coupons.
5. Pseudomonas is well known corrosive bacteria, it will develop biofilm, even though Klebsiella is very less biofilm end of incubation, other bacteria may influence inhibit the Klebsiella growth?
6. Which enzyme were presented in Biofilm?
Reviewer 2 Report
I found this to be a very interesting paper - well-written and clearly presented. Although I was of course aware that eDNA can be a significant contributor to the structure of bacterial EPS, I was surprised that it could contribute ">90% to the total biofilm composition", and it is on this point that I have my only major query: I suspect that the authors mean 90% of detected fluorescence, but how is this related to composition? Do any fluorescence standards exist so that fluorescence can be related directly to composition by volume or mass, for example? I don't understand what 90% of biofilm composition means, at present. Perhaps this could be described in the methods or supplementary materials.
Other than this, I have no major remarks and just a few small errors for correction:
Line 12 - Remove the ; after Biocides and replace with a comma. Remove the following comma.
Line 44 - I don't think 'sensible' is the correct word here.
Line 109 - : after setup.
Overall the paper does not contribute any truly remarkable results (although the results are certainly interesting), however in my opinion its major contribution is to document robust methods for carrying out such studies, which could be very useful to others in the field.
Reviewer 3 Report
This paper constitutes an interesting, and well written study of the EPS composition and community dynamics of biofilms growing in steel surfaces. In particular, the authors focused on the still quite undisclosed role of eDNA on biofilm development. The authors employed state- of the art approaches in one lab experimental setup, using DNA and RNA sequencing as well microscopic analysis and quantification of DNA. The results demonstrate the importance of eDNA for the development of marine biofilms, and their temporal dynamics, and this knowledge will have relevance for the biofouling and corrosion associated to bacterial biofilm formation on artificial surfaces. The objectives of this study are stated clearly in the manuscript, and the methodological section follows all the correct and most recent technological approaches in the topic, with a comprehensive methodology. The figures of the manuscript highlight appropriately the main findings of the work, and the discussion is well supported by the obtained results and the relevance of the results is made clear to the reader. As for the quality of writing, the English language quality is appropriate and the content is presented in a logical manner. The manuscript includes a decent number of references of previous work of the specific scientific area, however some sections, mainly in the introduction section, lack occasionally some bibliographical references. This manuscript suffers from the lack of some minor methodological details, and there is the need to rewrite some sentences for clarity purposes. Also, some figures were mentioned in the text but were not added to the manuscript. Other minor comments are included in the attached list of comments.
